On the frontiers of Twitter data and sentiment analysis in election prediction: a review

Alvi Quratulain 1
Ali Syed Farooq 1
Ahmed Sheikh Bilal 1
Khan Nadeem Ahmad 2
Javed Mazhar 1
Nobanee Haitham haitham.nobanee@liverpool.ac.uk 3 4 5
1 Department of Software Engineering, University of Management and Technology , Lahore , Punjab , Pakistan
2 Syed Babar Ali School of Science and Engineering, Lahore University of Management Sciences , Lahore , Punjab , Pakistan
3 Faculty of Humanities and Social Sciences, University of Liverpool , Liverpool , United Kingdom
4 College of Business, Abu Dhabi University , Abu Dhabi , United Arab Emirates
5 Oxford Centre for Islamic Studies, University of Oxford , Oxford , United Kingdom
Alatas Bilal
Electronic publication date: 2023 Aug 21
Publication date: 2023
Volume: 9
Electronic Location ID: e1517
Received 2023 Mar 15; Accepted 2023 Jul 14
Copyright: ©2023 Alvi et al.
Copyright year: 2023
Copyright holder: Alvi et al.
License: This is an open access article distributed under the terms of the Creative Commons Attribution License, which permits unrestricted use, distribution, reproduction and adaptation in any medium and for any purpose provided that it is properly attributed. For attribution, the original author(s), title, publication source (PeerJ Computer Science) and either DOI or URL of the article must be cited.
License URL: https://creativecommons.org/licenses/by/4.0/

Keywords: Sentiment Analsysi, Election prediction, Social media anlysis, Machine Learning, Policies, Classification, Social Media, Deep Learning, Twitter, Literature Review

Funding: The authors received no funding for this work.

==============================
Election prediction using sentiment analysis is a rapidly growing field that utilizes natural language processing and machine learning techniques to predict the outcome of political elections by analyzing the sentiment of online conversations and news articles. Sentiment analysis, or opinion mining, involves using text analysis to identify and extract subjective information from text data sources. In the context of election prediction, sentiment analysis can be used to gauge public opinion and predict the likely winner of an election. Significant progress has been made in election prediction in the last two decades. Yet, it becomes easier to have its comprehensive view if it has been appropriately classified approach-wise, citation-wise, and technology-wise. The main objective of this article is to examine and consolidate the progress made in research about election prediction using Twitter data. The aim is to provide a comprehensive overview of the current state-of-the-art practices in this field while identifying potential avenues for further research and exploration.

Introduction

The emergence of the information revolution has led to new economies centered around the flow of data, information, and knowledge (Serrat & Serrat, 2017). The Internet has brought about a significant transformation in content consumption. The vast amounts of data generated, coupled with the rapid dissemination of information and its easy accessibility, have turned social platforms into prime examples of interactions among millions of individuals (Gadek et al., 2017). These individuals actively engage with the shared content, effectively transforming their networks into successful platforms for exchanging information (Hu & Wang, 2020). Social media (SM) started its journey in the late 90s but got the world’s attention by providing a means of communication with people who are far away or to make friends. This ease became an addiction as it grew with more and more social networking sites.

Social networking sites allow people to express their thoughts, ideas, opinions, and feelings on various worldly and social matters through reactions, commenting, or sharing posts (Ceron, Curini & Iacus, 2015b). The exponential development of internet-based life and informal organization locales like Facebook and Twitter has begun to assume a developing part on certifiable legislative issues in recent years (Cottle, 2011).

Facebook and Twitter played a facilitating role for individuals, industries, and political nations worldwide (Segerberg & Bennett, 2011; Liao et al., 2018). Political parties such as Swedish Pirate Party, German Pirate Party, and Italy’s Five Star Movement Party used social networking sites to send the agenda to the whole country (Metzgar & Maruggi, 2009).

The USA election campaigns in 2008, 2012, and 2016 demonstrated the ground-breaking effect of SM on the general population of the United States. Obama was the first politician to effectively utilize SM as a campaign strategy (Smith, 2009). By the end, they knew the names of every one of the 69,406,897 citizens who were ready to vote in favor of Obama. To persuade the citizens, Obama hired an IT specialist in data mining and machine learning who sent customized messages for a cost-effective outreach to voters (Vitak et al., 2011). The overall digital enthusiasm for Trump was three times higher than Clinton, as indicated by Google Pattern Analysis, which made him victorious in the elections (MLLC, 2015). Donald Trump was the most mentioned person on Twitter and Facebook, with over 4 million Twitter followers more than Clinton (Stromer-Galley, 2014).

Following the footsteps, many other countries like Sweden (Larsson & Moe, 2012; Strömbäck & Dimitrova, 2011), India (Rajput, 2014; Pal, Chandra & Vydiswaran, 2016) and Pakistan (Ahmed & Skoric, 2014; Razzaq, Qamar & Bilal, 2014) also made extensive use of SM and conducted successful campaigns in the history of recent politics. The research community used different data analysis and mining processes and found the hidden patterns from trillions of data gathered from SM. They analyzed user sentiment from the written text on a user’s profile. This behavioral study is called sentiment analysis (SA) (Carlisle & Patton, 2013). Numerous election prediction algorithms were conducted using Twitter data based on sentiment analysis (Carlisle & Patton, 2013; Rajput, 2014), and the rest are discussed in the later sections.

After the USA elections (2008, 2012, 2016) and the Pakistan elections in 2013, the role of social media in politics, based on sentiment analysis, has been widely studied and examined (Carlisle & Patton, 2013; Wolfsfeld, Segev & Sheafer, 2013; Ahmed & Skoric, 2014; Razzaq, Qamar & Bilal, 2014; Safdar et al., 2015). During the research, a lot of election prediction was performed using Twitter data based on sentiment analysis (He et al., 2019; Ahmed & Skoric, 2014; Razzaq, Qamar & Bilal, 2014; Bagheri & Islam, 2017; Wang et al., 2012; Younus et al., 2014; Kagan, Stevens & Subrahmanian, 2015; Nickerson & Rogers, 2014). Numerous studies explore the realm of social media prediction, opinion mining, and information network mining techniques to establish standardized approaches to assess the predictive capabilities and limitations associated with the information embedded within social media data (Cambria, 2016; Kreiss, 2016; Mahmood et al., 2013).

The primary motivation of this study is to contribute to the existing body of scientific literature on sentiment analysis by focusing on its application in election prediction using Twitter data. This study aims to delve deeper into aspects that may have received limited attention in previous works. Through a systematic, comprehensive, and detailed method, this review offers a fresh perspective on the causal factors influencing temporal sentiment analysis in social media to stimulate further discussions and considerations for enhancing future studies in this domain. Furthermore, an integral part of our work, which we plan to expand in future research, is a practical evaluation of the applicability and reproducibility of existing and upcoming techniques. While these approaches exhibit impressive results, their practical implementation can be challenging. By offering insights into their potential limitations, we aim to provide a realistic outlook for their utilization.

There are generally three main levels of sentiment analysis: document level, sentence level, and aspect level sentiment analysis. Document-level sentiment analysis involves analyzing the overall sentiment of a document, such as a blog post or news article. Sentence-level sentiment analysis involves analyzing the sentiment of individual sentences within a document, while aspect-level sentiment analysis involves analyzing the sentiment expressed towards a specific aspect or feature of an entity, such as the battery life of a smartphone. Significant progress has been made in election prediction in the last two decades. This survey paper aims to examine the use of sentiment analysis for predicting election outcomes. Furthermore, it will identify research gaps and propose future research directions. The structure of the article continues as follows: the Literature Review section provides a theoretical framework by conducting a literature review to support the study. The methodologies employed are outlined in the Methodology section. The Results focuses on discussing the primary insights and results derived from the study. Finally, the Conclusion concludes the document by summarizing the main findings, limitations and highlighting potential areas for future research.

Theoretical framework

The exploration of election prediction using Twitter data and sentiment analysis has yet to be thoroughly examined within academia, indicating a need for an extensive survey of existing research in this domain. While some surveys have been conducted in the literature, they primarily focus on utilizing various social media platforms for election predictions, while others may be outdated or lack comprehensive coverage of all aspects related to election prediction using Twitter data. A recent survey (Nayeem, Sachi & Kumar, 2023) has been done in this field where researchers presented the significant publications ever done to analyze election prediction using different social media platforms. Some articles have (Baydogan & Alatas, 2022; Chakarverti, 2023; Baydogan & Alatas, 2021a) focused on evaluating the performance of artificial intelligence-based algorithms for hate speech detection and presents a novel approach for automatically detecting online hate speech. Baydogan & Alatas (2021b) proposed the use of the Social Spider Optimization algorithm for sentiment analysis in social networks while Baydogan & Alatas (2018) explores the use of the Konstanz Information Miner (KNIME) platform for sentiment analysis in social networks. Another paper by Rodríguez-Ibánez et al. (2023) examined sentiment analysis’s existing methods and causal effects, particularly in domains like stock market value, politics, and cyberbullying in educational centers. The paper highlighted that the research efforts are not evenly distributed across fields, with more emphasis on marketing, politics, economics, health, etc. Yu & Kak (2012) surveyed the domains that can be predicted utilizing current social media data by presenting a comprehensive overview of the existing methods and data sources used in past papers to predict election results. Kwak & Cho (2018) presented a survey that explores the insights gained and limitations encountered when utilizing social media data. The paper further examined the approaches to overcome these limitations and proposed effective ways of using social media data to comprehend public opinion in electoral contexts. Bilal et al. (2019) presented a survey listing the methods and data sources used in past efforts to predict election outcomes. In Rousidis, Koukaras & Tjortjis (2020), the authors examined current and emerging areas of social media prediction since 2015, specifically focusing on the predictive models employed. It reviewed literature, statistical analysis, methods, algorithms, techniques, prediction accuracy, and challenges. But this paper concentrates on something other than a specific field like politics. In Skoric, Liu & Jaidka (2020), authors presented the results of a meta-analysis that examines the predictive capability of social media data using various data sources and prediction methods. The analysis revealed machine learning-based approaches outperform lexicon-based methods, and combining structural features with sentiment analysis yields the most accurate predictions. Kubin & Von Sikorski (2021) investigated the influence of social media on political polarization. The study highlighted a heavy emphasis on Twitter and American samples while noting a scarcity of research exploring how social media can reduce polarization. The work in Cano-Marin, Mora-Cantallops & Sánchez-Alonso (2023) provided an evaluation and classification of the predictive potential of Twitter. The paper identified gaps and opportunities in developing predictive applications of user-generated content on Twitter.

Methodology

A systematic literature review was conducted following a six-step guideline for management research (Drus & Khalid, 2019) such as formulating the research questions, identification of necessary criteria for the study, potentially relevant literature retrieval, analyzing the relevant information gathered from the literature and the results of the review were reported. The current study addresses the following two questions:

1. What approaches are proposed by the research community to analyze the role of SM especially Twitter in politics?

2. How can we divide the research done in this area into different time-based intervals (eras)?   What are the main strengths and weaknesses of each era?

What are the main strengths and weaknesses of each era?

Design

To address the research inquiries, we conducted a systematic literature review (SLR) following the guidelines provided by Kitchenham (2004) and used it in many surveys in different fields. These guidelines emphasized the importance of identifying the need for the review, determining the relevant data sources, providing a comprehensive review process description, presenting the results clearly, and identifying research gaps to facilitate further investigation. To ensure the inclusion of recent and up-to-date methodologies employed by researchers, we collected a substantial corpus of 250 documents spanning from 2008 (after the launch of Twitter in March 2006) to March 2023.

To curate our dataset, we utilized multiple databases to filter publications based on publication dates. We extracted papers from the first three pages of the search results, ensuring a well-balanced dataset by prioritizing the most cited publications. We only selected the research papers, not surveys or reports. Through this SLR, we successfully analyzed 80 papers that conformed to our predefined criteria. The detailed stages are explained below. Figure 1 exhibits the visual representation of the methodology.

Figure 1 Methodology flowchart.

Stage 1: Screening

We collected 250 articles focusing on the elections of the USA 2008, Arab Spring 2010, USA 2012, Pakistan 2013, India 2014, USA 2016, and Pakistan 2018 from various databases such as IEEE, Springer, Emerald Insight, Science Direct, Scopus, and Association for Computing Machinery (ACM). The search criteria were based on keywords such as sentiment analysis, predicting election results and election prediction classification using social media, election prediction using sentiment analysis, election prediction using Twitter data, sentiment analysis using Twitter data, and social network analysis through sentiment analysis. The resultant articles were then analyzed based on the title and abstract of the articles. After the analysis, only those papers that directly correlated with election prediction and had valid digital object identifiers (DOI) were selected. In doing so, 162 articles were selected by the end of the screening process.

Stage 2: Eligibility analysis

After the screening process, publications related to analysis performed with other data sets than Twitter, like Facebook, surveys, and papers whose purpose was not to use Twitter as a predictive system, were excluded from our repository. So, the final number of eligible articles was further narrowed down to 80 papers as the study was based on election prediction using Twitter data. Thus, the resultant repository was quite reasonable for making conclusions and inferences about the impact of SM and different classification approaches performed on elections in various countries.

Pre-processing techniques

Pre-processing techniques are those technique(s) applied to the raw dataset to avail formatted, error-free dataset. The relevant algorithm(s) then use this processed dataset to achieve maximum accuracy with minimal deterioration in their otherwise smooth performance.

Stemming

Stemming is a technique that refers to all variations of a word to its root word, such as ‘warming’, ‘warmest’, ‘warmed,’ and ‘warmer’, which will be stemmed from the word ‘warm.’ This method reduces the time and memory space by removing suffixes that have exactly matching meanings and stem. For sentiment analysis on the text data, every word should be represented by the stem rather than the word mentioned in the text (Al-Khafaji & Habeeb, 2017).

Stop word removal

Stop words are the words that are useless within the raw dataset. These words do not provide helpful information in the data set, so they must be removed to save computation time, storage, and space and improve the algorithm’s efficiency. Most stop words are pronouns and helping verbs like is, of, the, to, and/or (Al-Khafaji & Habeeb, 2017).

Tokenization

Tokenization is a method to split the words within a sentence. Each split character, word, or symbol is called a token. It is an appropriate method in text analysis (Al-Khafaji & Habeeb, 2017). Like, [the president has worked well] will be tokenized into [the, president, has, worked, well] (Wongkar & Angdresey, 2019). These tokens help identify a content’s intent which helps in sentiment or text analysis.

Election prediction approaches

This section classifies all the research papers into various approaches. The taxonomy of these approaches is presented in Fig. 2.

Figure 2 Taxonomy tree of the approaches for election prediction.

Statistical approach

The authors of Ibrahim et al. (2015) presented a new approach for predicting the Indonesian Presidential elections in 2014. Their approach collected data from Twitter and preprocessed it by removing usernames and website links. Furthermore, the Twitter buzzers (Ibrahim et al., 2015) were eliminated using an automatic technique to collect data from real Twitter users and avoid unusual noise in it. The cleaned data was then subdivided into sub-tweets, labeled with a candidate’s name, and their sentiment polarity was computed. They further used mean absolute error (MAE) metric to evaluate the performance, resulting in an MAE of up to 61%.

In another study, Bansal & Srivastava (2018) introduced a novel method called Hybrid Topic Based Sentiment Analysis (HTBSA) for forecasting election results using tweets. The tweets were preprocessed using text formatting techniques, and then the topics were generated using the Biterm Topic Model (BTM). HTBSA was conducted based on the sentiments of topics and tweets, resulting in an 8.4% MAE (Eq. (1)). (1) MAE= ∑i=1n|yi−xi|/n.

Similarly, the authors presented Lexicon-based Twitter sentiment analysis for forecasting elections using emoji and N-gram features in 2019 (Bansal & Srivastava, 2019). Unlike previous studies, sentiment polarity will be analyzed using non-textual data such as emoji(s). The authors gathered data from Twitter while restricting themselves to Uttar Pradesh (UP) geo-location. The data was cleaned from HTML tags, scripts, advertisements, stopwords, punctuations, special symbols, and white spaces. Duplicate tweets were also eliminated in the process. The refined data was then converted to bi-grams and tri-grams, followed by sentiment labeling. Simultaneously, emoji unicode was compared with developed n-grams, and its sentiment was labeled. Consequently, both sentiments were used to calculate election prediction. Another mathematical algorithm was presented in (Nawaz et al., 2022) based on sentiment forecasting for Pakistan democratic elections. The authors manually annotated tweets to avoid implications of spammed data among the datasets. Then, aspects of filtered tweets were extracted, assigning grammatical forms to each word in the sentence. The gathered factors were associated with opinions using the semantic similarity measure RhymeZone (Whitford, 2014). Once the association was done, the Bayesian theorem was applied, which classified tweets with 95% accuracy.

Ontology approach

The authors of Budiharto & Meiliana (2018) forecasted the Indonesian presidential election using tweets from presidential candidates of Indonesia based on a preprocessing algorithm. The tweets were processed with text formatting techniques, including stopwords in the Indonesian language and special character elimination. Once the tweets were refined, top words, favorite lines, and re-tweets were counted. Then the authors calculated the polarity of positive, negative, and neutral reviews. In Salari et al. (2018), researchers proposed text and metadata analysis to predict Iran’s presidential elections in 2017. The text data were gathered in the Persian language from two different platforms: Telegram and Twitter messages. The data was then analyzed using various analyses: sentiment analysis of hashtags, sentiment analysis of posts using Lexipers (Sabeti et al., 2019), time analysis, and several views and users of each message analysis (Telegram). The first two analyses were text analysis, while the others were metadata information analysis. In doing so, the model achieved 97.3% accuracy in predicting the presidential election.

Lexicon based approach

In 2019, Oyebode and Oriji conducted sentiment analysis to forecast Nigeria’s presidential election 2019 (Oyebode & Orji, 2019). The data was extracted from Nairaland (Nelson, Loto & Omojola, 2018) using a web scraping approach, and they were preprocessed with text cleaning techniques. The resultant data were fed to three lexicon-based classifiers (Vader (Hutto & Gilbert, 2014), TextBlob, and Vader-Ext) and to train five machine learning classifiers, namely support vector machine (SVM), logistic regression (LR), multinomial naive Bayes (MNB), stochastic gradient descent (SGD), and random forest (RF). When the classifiers were evaluated, the proposed Vader-Ext outperformed all other classifiers as it resulted in an 81.6% accuracy rate.

Supervised learning approach

Machine learning (ML) is a field of software engineering that uses measurable procedures to enable computer systems to “learn” with information without being programmed explicitly. ML tasks are classified as supervised, unsupervised and deep learning. In 2010, the authors presented an automated method that evaluates sentiments via linguistically analyzed documents (Pak & Paroubek, 2010a). Those documents were trained for the NB classifier and tested using n-gram as a feature. In 2012, the authors of Wang et al. (2012) proposed a real-time election prediction system that analyzed the opinions of various users on Twitter. The opinions were anatomized and later used to train and test the NB classifier. A unique prediction model for Elections held in Pakistan in 2013 was presented in Mahmood et al. (2013). A set of tweets were gathered according to predictive models, which were later cleaned and were used to train CHAID (chi-squared automatic interaction detector) decision tree (DT), SVM, and NB classifiers. When the classifiers were evaluated with test data, the CHAID decision tree dominated compared to SVM and NB (Eq. (2)). (2) χ2= ∑Oi−Ei2Eiwhereχ2=chisquared,Oi=observed value,Ei=expected value.

In 2014, the authors of Razzaq, Qamar & Bilal (2014) proposed a prediction system that evaluated the power of election prediction on the Twitter platform. The authors gathered and preprocessed tweets by eliminating duplicate tweets, URLs, whitespaces, and manual labeling. Furthermore, the Laplace method and Porter Stemming avoided zero values. Processed training data was used to train RF, SVM, and NB classifiers. When the classifiers were tested, NB dominated SVM and RF. Jose & Chooralil (2015) introduced a novel election prediction model using word sense disambiguation. The data was acquired from Twitter which was then cleansed by removing usernames, hashtags, and special characters. The negation handling technique was also applied to enhance the classification accuracy further. Speech tagging and tokenization were done on refined tweets provided by a word sense disambiguation classifier for categorization. The classifier attained a 78.6% accuracy rate. (3) ∼F1=2∗TNTN+FN∗TNTN+FPTNTN+FN+TNTN+FP

Tunggawan & Soelistio (2016) presented a predictive model for the 2016 US presidential election. They gathered data from Twitter, which went through simple filtration techniques such as URLs and candidate name removal to make the resultant data precise. In doing so, 41% of the data were eliminated. Then the data was labeled manually and fed to the NB classifier (Eq. (3)). The classifier predicted 54.8% accuracy. Sharma and Moh proposed a supervised election prediction method using sentiment analysis on Hindi Twitter data (Sharma & Moh, 2016). In this method, raw Hindi Twitter data underwent a text-cleaning module that removed negated words, stopwords, special characters, emoticons, hashtags, website URLs, and retweet text. 2 supervised (NB, SVM) and one unsupervised (Dictionary based) algorithms were used for classification. The dictionary-based classifier evaluated the tweets with 34% accuracy, whereas NB and SVM classifiers were trained with 80% accuracy of the data. In contrast, the remaining 20% of the data was used for the evaluation purpose, which resulted in 62.1% and 78.4%, respectively.

Ramteke et al. (2016b) presented a two-stage election prediction framework using sentiment analysis using Twitter data and TF-IDF. Further, the data was labeled using hashtag clustering and VADER techniques. 80% of the labeled data was used for training, and the remaining 20% was used for testing the classification algorithm. An accuracy rate of 97% was achieved when the classifier was tested. Ceron-Guzman and Leon-Guzman presented a sentiment analysis approach on Colombia Presidential Election 2014 (Cerón-Guzmán & León-Guzmán, 2016). Twitter data was cleaned and normalized in two stages: basic and advanced pre-processing. The data was stripped from URLs, emails, emoticons, hashtags, and special characters in the basic pre-processing stage. After then, the data was forwarded to the advanced pre-processing step, where lexical normalization and negation handling techniques were applied to refine the data further. Once the text was normalized, it was modified to a feature vector, which was later fed to the classifier. The labeled dataset was split into 80:20 ratios for training and testing classifiers. Overall, the classifier performed with 60.19% accuracy when evaluated on test data. Singh, Sawhney & Kahlon (2017) presented a novel method for forecasting US Presidential elections using sentiment analysis. After collecting data from Twitter, the authors implied a restriction to consider only one tweet per user. All duplicate tweets were removed to avoid interference affecting the method’s performance. Then unwanted HTML tags, web links, and special characters/symbols were removed from the data. The refined data was then used to train the SVM classifier. Once the classifier was trained, it was evaluated with the test data, i.e., to classify the polarity of the data, attaining a 79.3% accuracy rate.

In 2018, Bilal et al. (2018) presented a deep neural network application to forecast the electoral results of Pakistan 2018. They collected 56,000 tweets about the general elections in 2013 and treated them with text-cleaning techniques. The resultant data was then used to train Recurrent Neural Network (RNN). Once the RNN was trained, it was evaluated with test data resulting in an 86.1% prediction rate. In 2019, a new methodology was presented in Joseph (2019), which predicted Indian general elections using a decision tree. Ruling and opposing parties’ data was gathered from Twitter. Stopwords, regular expressions, emojis, Unicode, and punctuation, were pruned from the data. The resultant data was then tokenized and fed to the decision tree classifier for training. Once the classifier was trained, it was evaluated, which resulted in 97.3% accuracy. An efficient method to forecast the Indonesian presidential election using sentiment analysis was presented in Kristiyanti & Umam (2019). The authors collected data from Twitter, tokenized them, and generated Bi-grams(three-letter word combinations). Unlike previous research, the feature selection was made using the particle swarm optimization (PSO) algorithm and genetic algorithm (GA) algorithm separately. Then those features were used to train the SVM classifier. Once the classifier was trained, SVM with PSO performance dominated, against SVM with GA, with 86.2% accuracy rate.

Similarly, Oussous, Boulouard & Zahra (2022) proposed another Arabic sentiment analysis framework that forecasted the Moroccan general election 2021. The data was collected in Arabic from the Hespress website (a Moroccan news website). Then it was treated with text cleaning techniques (tokenization, normalization, and stop words removal) so that the dimensionality and processing time of the framework could be reduced. Term frequency (TF) was used to acquire feature vectors which were then passed on to several ML classification models such as SVM, NB, Adaboost, and LR for training. The classifiers predicted sentiment polarity with 94.35%, 62.02%, 87.55%, and 88.64% accuracy rates.

Deep learning approach

Hidayatullah, Cahyaningtyas & Hakim (2021) conducted sentiment analysis using a neural network to predict the Indonesian presidential election 2019. Two different datasets, i.e., before and after the elections, were collected and labeled using a pseudo-labeling technique. Then they were preprocessed with text-cleaning techniques, including case folding, word normalization, and stemming. This study trained three traditional ML classifiers (SVM, LR, and MNB) and five deep learning classifiers (LSTM, CNN, CNN+LSTM, GRU+LSTM, and bidirectional LSTM). Once the classifiers were ready, they were all evaluated by test data. SVM and bidirectional LSTM ruled better accuracy within their respective categories, but overall, bidirectional LSTM outperformed SVM by gaining an 84.6% accuracy rate. In another study, researchers presented a method for predicting USA presidential elections 2020 using social media activities (Singh et al., 2021). The dataset was pretreated with text formatting techniques plus TF-IDF vectorization. They were then fed into NB, SVM, TextBlob, Vader, and BERT classification models for training and testing purposes. Compared to other classifiers, the BERT classifier prevailed with a 94% precision rate.

In Ali et al. (2022), the authors introduced a deep learning model to forecast Pakistan general elections based on sentiment analysis. The dataset related to Pakistan general elections 2018 was gathered from Twitter, and they were labelled manually. Then the data was preprocessed with data transformation, tokenization, and stemming. Further, the proposed deep learning model was trained and evaluated with training and test data, respectively, resulting in a 92.47% prediction rate.

Research trends in algorithms and techniques of SM in politics: from beginning to date

Research objective 2: What are the research trends in algorithms and techniques of big social data in different time-based eras?

Era 1 (2010–2017)

This was the era where the research community were onset on innovating and developing election forecasting algorithms before actual election commencement. The prompt goal of their work was to predict and classify sentiments among digital text with optimal accuracy rate.

Pak & Paroubek (2010a) presented a novel method to forecast elections using sentiment analysis. Thus, the method predicted with 60% accuracy. In 2012, Wang et al. (2012) proposed an election prediction system focusing on US Presidential election 2012 data. Unigram features were extracted from 17,000 tweets (training dataset) and were fed to the NB classifier for training. Once the classifier was trained, it was evaluated with a 59% prediction rate. Mahmood et al. (2013) proposed an election prediction method that forecasted Pakistan General Election 2013 by assessing the CHAID decision tree. In doing so, the classifier performed with a 90% prediction rate. Razzaq, Qamar & Bilal (2014) presented a machine learning algorithm that predicted positive and negative sentiments with 70% accuracy. Thus, this method needed to be more consistent due to a biased data set. Ibrahim et al. (2015) proposed a statistical prediction method enthralled on Indonesian Presidential elections 2014. The method performed with 0.61 mean absolute error (MAE). There were several limitations to this method. Firstly, a dataset of all Indonesian voters across all Indonesian provinces should have been considered. Secondly, sentiment analysis (SA) cannot be performed when no keyword is present in candidate-related tweets. In 2015, Jose & Chooralil (2015) proposed an election prediction method using word sense disambiguation. Although the method performed with a 78.6% accuracy rate without the training phase, it was limited to negation handling and manual labeling.

In Tunggawan & Soelistio (2016), the authors innovated a Bayesian election prediction system focusing 2016 US presidential election. Although the system boomed with an exceptional accuracy rate with model test data, it under-performed with a 54.8% accuracy rate when evaluated with test poll data. Similarly, Sharma & Moh (2016) presented an Indian election prediction system using the Hindi dataset. The tweets were preprocessed, and then the polarity of the resultant tweets was calculated. SVM achieved a 78% prediction rate. The system is curtailed with emoticon analysis and extensive training data. A sentiment analysis system was introduced in Cerón-Guzmán & León-Guzmán (2016) predicting Columbia presidential election 2014. It resulted in the lowest MAE of about 4%. In Singh, Sawhney & Kahlon (2017), a sentiment analysis system was presented focusing on US presidential elections 2016. The system got trained with the Twitter processed data, and later, it was evaluated with test data, resulting in a 79% accuracy rate. A separate study (Ceron, Curini & Iacus, 2015a) examined the advantages of supervised aggregated sentiment analysis (SASA) on social media platforms to forecast election outcomes. Analyzing the voting intentions expressed by social media users during several elections in France, Italy, and the United States between 2011 and 2013, they compared 80 electoral forecasts generated through SASA alongside alternative data-mining and sentiment analysis approaches.

Era 2 (2018–2023)

This era is characterized as embarking of deep learning approach as it provided a major breakthrough and expedited state of the art results among its classification algorithms. It gave a new direction towards accuracy improvisation. In doing so, the research community focused on creating and developing new deep learning algorithms as compared to machine learning algorithms. Furthermore, this era also saw a novel sentiment classification proposals relying on statistical, lexicon and ontology approaches as seen in Fig. 3.

Figure 3 Distribution of various categories of approaches in Era 1 and Era 2.

Bilal et al. (2018) introduced a deep neural networks (DNN) election prediction model that forecasted Pakistan General Elections 2018 with an 86.1% accuracy rate. Case sensitive tweets of tweets deteriorated the performance of the method. Kristiyanti & Umam (2019) proposed a sentiment analysis method to predict the Indonesian presidential election for 2019–2024. The system utilized particle swarm optimization (PSO) and genetic algorithm (GA) algorithms with SVM to improvise accuracy to 86.2%. Salari et al. (2018) presented an election prediction system for Iran presidential election 2017. Both text and metadata analysis of the tweets were considered to evaluate the system’s performance. The system performed with a 97.3% accuracy rate without the training phase. Another outcome prediction system, based on Indian general elections, was presented by Joseph (2019), which trained and tested the DT classifier resulting in a 97% accuracy rate. Thus, this system works well with tweets in the English language only.

Chaudhry et al. (2021) proposed an election prediction method, mainly focusing on the US election 2020. They collected the Twitter data, preprocessed them, and extracted features using TF-IDF. Features of around 60% of the training dataset were used to train the NB classifier. In contrast, the features of the rest 40% dataset were utilized to evaluate the performance, resulting in a 94.58% accuracy rate. Thus, the authenticity of the dataset (tweets) was not examined, which hurt the method’s performance. Likewise, Xia, Yue & Liu (2021) proposed a sentiment analysis-based election prediction method for the same election campaign. The authors preprocessed the tweets with string replacement and stemming techniques in this method, followed by n-gram feature extraction. Multi-layer perceptron (MLP) classified 27,840 dimension features with an accuracy of 81.53%.

An election prediction method based on a deep learning approach was introduced in Hidayatullah, Cahyaningtyas & Hakim (2021), which forecasted the Indonesian Presidential elections in 2019. The authors trained CNN, long-short term memory (LSTM), gated recurrent unit (GRU), and bidirectional LSTM, SVM, LR, and multinomial NB, from which bidirectional LSTM dominated against the rest by achieving 84.6% prediction rate. It also implied that, in comparison, DNNs attained a better accuracy rate than traditional machine learning algorithms. Similarly, another deep learning approach was implemented to forecast US presidential elections 2020 in Singh et al. (2021). Three machine learning algorithms (SVM, NB, and TextBlob) and one deep learning algorithm (BERT) were trained and evaluated. As a result, the BERT algorithm attained the highest prediction rate of 94%. It denoted that DNN algorithms achieved better accuracy than conventional machine learning algorithms. Ali et al. (2022) introduced another DNN election prediction method focusing mainly on Pakistan General Elections 2018. The data were labeled manually, preprocessed, and later tokenized as usual. The resultant dataset was used to train and evaluate the DNN classifier, resulting in a 92.47% accuracy rate. Thus, the dataset used in the method above needed to be higher, due to which accuracy dropped. Previously, traditional polling data was widely considered the most reliable method for forecasting electoral outcomes. However, recent developments have revealed polling data’s potential incompleteness and inaccuracy. A study was conducted to compare the accuracy of polls with sentiment analysis results obtained from Twitter tweets (Anuta, Churchin & Luo, 2017). The study analyzed a new dataset of 234,697 Twitter tweets related to politics, collected using the Twitter streaming API. The tweets underwent preprocessing, removing hashtags, links, and account names and replacing emotions and symbols with their complete form. The study’s findings indicated that Twitter exhibited a 3.5% higher bias in popular votes and a 2.5% higher bias in state results compared to traditional polls. Consequently, the study concluded that predictions based on Twitter data were inferior to those found on polling data (Anuta, Churchin & Luo, 2017). The researchers highlighted the limitations of previous methods. They recommended incorporating additional techniques, such as POS tagging and sense disambiguation, during preprocessing and considering contextual and linguistic features of words to enhance prediction accuracy (Anuta, Churchin & Luo, 2017).

In the context of the 2016 US elections, traditional techniques like polling were deemed unreliable due to the rapid evolution of technology and the prevalence of social and digital media platforms (Hinch, 2018). A study analyzed slogans used in Twitter tweets during the elections, employing a WordCloud visualization. However, the analysis results could have been more consistent with the actual election outcomes, particularly in predicting Trump’s victory in Michigan and Wisconsin. The researchers emphasized the need to consider qualitative aspects when making electoral predictions, as the approaches employed in the study failed to capture the dynamics accurately.

The relationship between candidates’ social network size and their chances of winning elections was examined in a study that utilized data from Facebook and Twitter (Cameron, Barrett & Stewardson, 2016). The study employed regression analysis and proposed three models, with the number of votes as the dependent variable and the number of Facebook connections and other factors as independent variables. The results indicated a significant correlation between the size of the social network and the likelihood of winning. However, the effect size was small, suggesting that social media data is predictive only in elections with close competition.

A study used social network techniques, such as volumetric analysis and sentiment analysis, to infer electoral results for Pakistan, India, and Malaysia (Jaidka et al., 2019). The study collected approximately 3.4 million tweets using the Twitter streaming API and separated English tweets using a natural language toolkit. Volumetric analysis, measuring the volume of tweets for each party; sentiment analysis assessing positive and negative tweets; and social network analysis determining the centrality score of each party were employed. The study found that Twitter data was ineffective for making election predictions in Malaysia but proved effective and efficient for Pakistan and India. Incorporating multiple techniques, the proposed model was also effective for candidates and parties with fewer votes.

A study conducted in 2016 proposed a predictive model for forecasting the outcome of the US presidential elections based on an NB approach utilizing Twitter data (Tunggawan & Soelistio, 2016). The researchers collected tweets from December to February, covering three months. The collected data underwent simple pre-processing techniques to prepare it for sentiment analysis. The resulting model achieved an impressive accuracy of 95.8% in sentiment prediction. A 10-fold cross-validation technique was employed to assess the model’s robustness. The F1 Score was used to evaluate the model’s accuracy in predicting positive sentiments, while F1 represented the model’s accuracy in classifying negative sentiments. The model’s accuracy in predicting negative sentiments (Tunggawan & Soelistio, 2016). The authors of Heredia, Prusa & Khoshgoftaar (2018) introduced their sentiment analysis model, which classified the data with an accuracy of 98.5%. However, when the model’s predictions were compared with actual polling data, the results indicated an accuracy of 54.8%.

Findings

The statistical analysis of the included research papers revealed interesting insights regarding the distribution of publications across conferences and journals as shown in Fig. 4. The data indicated that a substantial portion of the research publications were disseminated through conferences, accounting for 64% of the total publications. On the other hand, research papers published in reputable journals constituted 9% of the overall distribution as can be seen in Fig. 4a. This finding highlights the significance of conferences as platforms for rapid knowledge sharing and the enduring impact of journals in disseminating scholarly research.

Figure 4 Statistical analysis of research papers. (A) Distribution of various research publications into conferences and journals. (B) Distribution of various research publication into different approaches.

The analysis further examined the distribution of research publications across different approaches. It was observed that a diverse range of approaches were employed across the reviewed papers. As seen in Fig. 4b, the data indicated that machine learning approach constituted the highest proportion accounting for 90% of the publications, followed by deep learning approach with 20%. This distribution showcases the varied methodologies utilized by researchers within the field and the prominence of certain approaches in contributing to the existing body of knowledge. To determine the years in which the authors exhibited a greater influence through their publications, an examination of publication trends from 2010 till 2022 was conducted and presented in Fig. 5.

Figure 5 The years in which the authors published with a greater influence.

The analysis revealed that the years 2015 and 2022 had the most publications in this area. Articles written in the year 2023 are not included as the numbers might change by the end of the year. This shows that trend is increasing for this domain. Moreover, the paper analyzed the distribution of various categories of approaches in era 1 and era 2, representing different time frames of papers published on the topic. The aim was to investigate the evolution and trends of research methodologies and approaches used in the field over time. Fig. 3 presents the distribution of approaches across era 1 and era 2.

The analysis depicted that era 1 was dominated with machine learning algorithms but the shift started changing from era 2 to deep learning approach. Overall, the distribution of approaches in the field has undergone changes between era 1 and era 2, indicating an evolving research landscape. The shift towards deep learning and lexicon approaches suggests a diversification of research methodologies and a broadening of research interests in the field over time. These findings highlight the importance of understanding the temporal dynamics in research approaches and methodologies within the topic, providing valuable insights into the progression and development of the field.

Furthermore, the analysis identified the most relevant and highly cited journals in this domain as shown in Table 1. Through a thorough examination of the citations within the reviewed papers, it was found that out of 42 journals, 15 contained articles with citations above a hundred. It was further seen that Journal of Social Science and Computer Review emerged as the most relevant and cited journal, followed by the journal, First Monday. These journals have consistently published influential research within the domain, indicating their significance as reputable outlets for disseminating scholarly work

Table 1 Most relevant and cited journals.

Journal	Number of Publications	Citations	
Cyber Psychology, Behavior, and Social Networking	1	906	
Mass Communication and Society	1	186	
Political Research Quarterly	1	294	
Journal of Broadcasting & Electronic Media	1	165	
The International Journal of Press/Politics	1	267	
First Monday	1	1023	
Association for Computational Linguistics	1	182	
Communications of the ACM	1	177	
Electoral Studies	1	263	
Journal of Big Data	3	297	
arXiv	3	257	
Journal of Ambient Intelligence and Humanized Computing	2	110	
Social Science Computer Review	2	1049	
European Journal of Communication	2	752	
IEEE Intelligent Systems	2	178	

Similarly, the analysis identified the most relevant and highly cited conferences in this domain as presented in Table 2. The data showed that most papers were submitted to conferences on the web and social media, artificial intelligence, and on conferences on big data. The meta-analysis in this paper further helped in compilation of ten most highly cited articles, serving as a means to identify publications of significant research interest in Table 3.

Table 2 Most relevant and cited conferences.

Conference	Number of Publications	Citations	
AAAI Conference on Web and Social Media	4	4023	
Conference on System Sciences	3	327	
Conference on data mining and advanced computing	2	107	
Proceedings of the workshop on semantic analysis in social media	1	376	
AAAI conference on artificial intelligence	1	225	
Conference on Language Resources and Evaluation	1	4135	
InProceedings of the ACL 2012 system demonstration	1	879	
Conference on big data	1	118	
Conference on inventive computation technologies	1	154	

Table 3 Top 10 most cited papers.

References	Year	Focus of Study	Citations	
Pak and Paroubek (2010b)	2010	Sentiment Analysis On Election Tweets	4135	
Tumasjan et al. (2010)	2010	Election Prediction With Twitter	3646	
Wang et al. (2012)	2012	Twitter Sentiment Analysis Of 2012 Us Presidential Election	879	
Tumasjan et al. (2011)	2011	Election Prediction With Twitter	660	
Sang and Bos (2012)	2012	2011 Dutch Election Prediction With Twitter	376	
Burnap et al. (2016)	2016	Twitter Sentiment Analysis Of 2015 Uk General Election	263	
Ramteke et al. (2016b)	2016	Election Prediction With Twitter	154	
Budiharto and Meiliana (2018)	2018	Twitter Sentiment Analysis Of Indonesia Presidential Election	147	
Ramteke et al. (2016a)	2016	Election Prediction From Twitter Using Sentiment Analysis	136	
Sharma and Moh (2016)	2016	Sentiment Analysis On Hindi Twitter	118	

The paper has drawn comparisons between the findings of this research and the most relevant works in academia within the field. These comparisons aimed to situate the current study within the existing literature and highlight its contributions. The results align with previous studies that emphasize the importance of conferences and journals in disseminating research findings. Additionally, the prevalence of specific approaches identified in this research aligns with prior works that have identified and discussed these approaches in the literature.

This study holds several implications from theoretical, managerial, and practical standpoints. Theoretical implications include further validating and expanding existing theories and frameworks within the field, particularly in relation to the distribution of research publications and the prevalence of different approaches. The findings of this study contribute to the overall understanding of the research landscape and can serve as a basis for future theoretical developments and investigations.

From a managerial perspective, the results offer insights into the most influential years and the distribution of research approaches. This knowledge can assist managers and decision-makers in understanding the trends and dynamics of the field, enabling them to make informed decisions regarding resource allocation, collaboration opportunities, and strategic planning.

Practically, this research provides valuable guidance for researchers and scholars in terms of selecting appropriate publication outlets and identifying the prevailing approaches in the field. The identification of the most relevant and cited journals and conferences can aid researchers in targeting their work for maximum impact and visibility. Furthermore, knowledge of the most cited papers within the domain helps researchers stay abreast of seminal works and establish connections with influential researchers.

Conclusion

This study provides a detailed analysis of existing sentiment classification techniques in chronological order and categorizes them into statistical, lexicon, oncology, supervised, unsupervised, and deep learning approaches. It can be concluded that deep learning approach produced promising results. Despite that, deep learning constitutes new challenges such as high computational requirements and large dataset for training its models. The review paper further addresses the existing gap in the literature on election prediction using sentiment analysis of Twitter data. It contributes to the field by thoroughly analyzing existing studies, evaluating the effectiveness of sentiment analysis as a predictive tool, identifying challenges associated with this approach, and discussing the implications and future directions for research. By consolidating the findings, highlighting limitations, and suggesting potential advancements, this review is a valuable resource for researchers, practitioners, and policymakers interested in utilizing sentiment analysis to predict election outcomes and understand public opinion.

It has been analyzed that while there may be observed correlations between specific Twitter trends or sentiment patterns and election outcomes, it does not necessarily imply that these correlations indicate a causal relationship or direct influence on the election results. Merely correlating Twitter data and election results does not mean that the sentiment expressed on Twitter is causing the election outcome. Other factors, including traditional polling data, campaign strategies, socioeconomic factors, and voter behavior, may play more significant roles in determining the election outcome. Integrating multiple data sources and carefully considering other relevant factors to address this limitation is crucial. By doing so, researchers can mitigate this limitation and achieve a more accurate and comprehensive understanding of the dynamics underlying elections.

Moving forward, there are several areas where sentiment analysis for election prediction can be further scrutinized to enhance the efficiency and accuracy of classification algorithms. Incorporating additional data sources such as news articles, television transcripts, and survey data can provide a more comprehensive view of public opinion and enable the development of robust models that mitigate bias within extensive training data. Furthermore, improving sentiment analysis models to encompass diverse source data and exploring various aspects of the text, including sarcasm, subjectivity, and emotions, can contribute to predicting sentiment with higher precision.

We would like to extend our gratitude to Sheikh Bilal Ahmed for his assistance. We are grateful to all the anonymous reviewers for their useful comments.

Additional Information and Declarations

Competing Interests

Author Contributions

The authors declare there are no competing interests.

Quratulain Alvi conceived and designed the experiments, performed the experiments, analyzed the data, prepared figures and/or tables, authored or reviewed drafts of the article, and approved the final draft.

Syed Farooq Ali conceived and designed the experiments, analyzed the data, authored or reviewed drafts of the article, and approved the final draft.

Sheikh Bilal Ahmed conceived and designed the experiments, performed the experiments, analyzed the data, prepared figures and/or tables, authored or reviewed drafts of the article, and approved the final draft.

Nadeem Ahmad Khan analyzed the data, prepared figures and/or tables, and approved the final draft.

Mazhar Javed conceived and designed the experiments, authored or reviewed drafts of the article, review, and approved the final draft.

Haitham Nobanee performed the experiments, authored or reviewed drafts of the article, review, and approved the final draft.

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
