# Peer review of "On the frontiers of Twitter data and sentiment analysis in election prediction: a review"

_PeerJ Computer Science, doi:10.7717/peerj-cs.1517_

## Round 0.1 · original submission · Major Revisions

Dear authors,

Your article has not been recommended for publication in its current form. However, we do encourage you to address the concerns and criticisms of the reviewers and resubmit your article once you have updated it accordingly.

Reviewer 1 has requested that you cite specific references. You may add them if you believe they are especially relevant. However, I do not expect you to include these citations, and if you do not include them, this will not influence my decision.

Reviewer 1 ·

Basic reporting

Outline

First of all, I would like to congratulate the author(s) for their work. Please be assured that I will evaluate the study only from a scientific point of view, without prejudice.

This work is valuable and prudent work in the computer science community and social media area. Also, I can't end without saying that it is worth reading.

In the study, the author(s) survey paper aims to provide an entry point for all interested researchers and data mining experts to learn key algorithms, techniques, and growing innovations for election prediction.

Also, using the powerful lens of literature, the current study reviews the main approaches in the field, discusses trends in different eras, discussion of the strengths and limitations of these methods, and summary of the essential findings and future directions for research in this area.

Experimental design

Design and Writing
The article should be prepared by considering the journal template writing rules (see: https://peerj.com/about/author-instructions/cs).
Although the article was prepared with great effort, there are typographical errors in the most prominent parts of the article. These deficiencies can't be tolerated for the valuable study.

- The article should be completely revised from the use of abbreviations to reference notation.

ex1. SNS are social media platforms where people are empowered to express their thoughts, ideas, opinions, feelings, etc. on various worldly and social matters via reactions, commenting or sharing posts in their news feed [13]. The exponential development of internet-based life and informal organization locales like 39 FB and Twitter has begun to assume a developing part on certifiable legislative issues in recent years [16]. FB and Twitter played a facilitating role for individuals, industries, and political nations worldwide [49, 30]. SNS has been used, for example, to orchestrate uprisings amid ’Arab spring’ [17].

- Also, choose correctly the verbs that emphasize the achievement results of the literature.

ex2: Similarly, Pirate Party in Sweden and Germany and the ’Italian Movimento 5 Stelle’ in Italy used SNS to attract individuals and share the mandate of their party at regular intervals and achieved a HUGE SUCCESS in sending their voice to the whole country [33].

- Attention should be paid to the use of abbreviations.

ex3: This ease became an addiction as it grows with more and more SNS. (What is SNS?). After ’Arab Spring’, USA Elections (2008, 2012, 2016), and Pakistan elections in 2013, SM role in politics based on sentiment analysis has been widely studied and examined [12, 60, 3, 44, 47].It was seen during the analysis that a lot of election prediction was performed using Twitter data based on sentiment analysis [18, 3, 44, 6, 57, 11, 27, 31, 63, 26, 37].

- Some of your sentences should be shortened and clarified.

ex4. Methodologies used in SA for election prediction, such as text mining, deep learning and machine learning and how they have been applied to election prediction.

- Incomplete and inconsistent sentences should be completed. In addition, the article should be cleared of all grammatical errors by using the grammar check tool (For example, Grammerly or Ginger).

Validity of the findings

In the abstract section, clearly emphasize your main motivation for the preparation of the article. The last sentence of the Abstract section must have been completed with a more striking sentence.

In the conclusion section, list your achievements with this study. Clearly express your contribution to future studies and literature.

Satisfactory technical information about the method used in the literature is not presented. No mathematical expressions, from the evaluation metrics used to the algorithms, are included in the article.

Please clearly describe the methods and algorithms analyzed.

Not only the methods and methods applied for election, but also those used in other SA studies should be investigated. Attention should be paid to machine learning deep learning, and optimization methods. Please check the additional comments section.

Provide the motivation for the conclusion part.

Additional comments

In addition, in order to examine the latest technology artificial intelligence and optimization algorithms, read the following articles and add all that seem relevant to your article.

2022, Deep-Cov19-Hate: A Textual-Based Novel Approach for Automatic Detection of Hate Speech in Online Social Networks throughout COVID-19 with Shallow and Deep Learning Models

2021, Performance Assessment of Artificial Intelligence-Based Algorithms for Hate Speech Detection in Online Social Networks https://doi.org/10.35234/fumbd.986500

2021, Metaheuristic Ant Lion and Moth Flame Optimization based Novel Approach for Automatic Detection of Hate Speech in Online Social Networks

2021, Sentiment Analysis in Social Networks Using Social Spider Optimization Algorithm

2019, Detection of Customer Satisfaction on Unbalanced and Multi-Class Data Using Machine Learning Algorithms

2018, Sentiment analysis using Konstanz Information Miner in social networks

Reviewer 2 ·

Basic reporting

- In order to ensure clarity and organization in your article, I kindly request that you follow the IMRAD framework structure, which includes Introduction, Literature Review/Theoretical Framework, Methods, Results, Discussion and Conclusion sections. This framework will help readers to easily understand your research and findings, and is a widely accepted structure for scientific articles.
- Improve storytelling and link between paragraphs in the Introduction as it seems there is a collection of unlinked ideas. What is the motivation of this research? It is recommended to rewrite this section.
- A new section "Theoretical framework" should be created. It should review the main works in the domain and should comment on the main contributions of articles which followed a similar methodology.
- Ensure the formulated research questions are supported by the existing literature.
- I recommend review the existing literature in this research domain and in this journal and review the most cited documents in the same area. As a result, you would ensure the alignment of this manuscript with the editorial line set by this journal. For example:

Kubin, E., & von Sikorski, C. (2021). The role of (social) media in political polarization: a systematic review. Annals of the International Communication Association, 45(3), 188-206.
Cano-Marin, E., Mora-Cantallops, M., & Sánchez-Alonso, S. (2023). Twitter as a predictive system: A systematic literature review. Journal of Business Research, 157, 113561.
Rodríguez-Ibánez, M., Casánez-Ventura, A., Castejón-Mateos, F., & Cuenca-Jiménez, P. M. (2023). A Review on Sentiment Analysis from Social Media Platforms. Expert Systems with Applications, 119862.

- I recommend to group the sections 3-6 in a new section called Methodology.
- It is recommended to add a flowchart that summarises the methodology implemented.
- A specific section for Results should be added, giving answer to the main research questions. Creating a specific section for Results in a paper is important because it allows the reader to easily locate and understand the key findings of the study.
- I would recommend that the authors professionally proofread the paper carefully to ensure that it is free of language and formatting errors. You should ensure consistency between British and American English (e.g. behavioral vs behaviour). This will improve the clarity and readability of the paper and help to ensure that the authors' ideas and findings are presented in the best possible light.
- The format of the references should be checked. Review the format of the references and citations to ensure consistency. In addition, reference number [2] is empty.

- The format of the tables should be more professional, aligned with the format used in the published articles in the journal.

- Table 1 should be an appendix.
- Figure 1: the time axis is not readable. Please, recreate it and prevent the use of logos that reprensent trademarks.

Experimental design

- The methodology should clearly present the process to collect data as this impacts on the repeatibility of the study. Clarify the time horizon of the research.
- Why are you only analysing 55 scientific articles? Please, include this as a limitation and justify the inclusion and exclusion criteria.
- Comment on the limitation about spurious correlation and correlation does not imply causation.
- Figure 3/Figure 4: Avoid at any cost the use of pie charts and replace those by a bar chart.
- Table 2: what is the value that this table adds? Please, remove. Same in the case of Table 3.

Validity of the findings

- A discussion section should be created, as recommended before. It provides an opportunity to reflect on the implications of the research findings and to contextualize them within the broader literature. A well-written discussion can help the reader to understand the significance of the study and its contribution to the field.

- What was the existing gap and what are the main contributions of this research to the existing literature? These are the questions to answer in the Conclusion. It is important to strengthen and further work on this section as it should summarise the value creation originated in this study.

·

Basic reporting

The manuscript discusses the computational and algorithmic developments in the field of election result prediction using tweets in the last two decades. I would congratulate authors for a timely research problem which will be useful to many potential researchers. The writing is crisp and almost unambiguous. The authors provided the references contextually.
This is a systematic review work, leading to a cross-disciplinary interest. Although few such studies are published recently, yet this work focuses on the election prediction, which is needed as an focused study. However, the presented work in this work does not start with a sound and acceptable norms in the systematic review domain. For example, the two research questions are overlapped. The databases are not exhaustive. Some of these points are discussed in detail in the next section on study design.

Experimental design

Systematic review needs to define the keywords carefully. The paper by Drus and Khalid (2019) on "Sentiment Analysis in Social Media and Its Application: Systematic Literature Review", specifies the study design one should use in this kind of systematic review works.
First of all, the research questions should be formulated, as both exclusive and exhaustive, keeping the focus on the research objectives. In this paper, the research questions are not exclusive. Next, the key-words used should be define a priori and must be reflective of the research questions, Here, the keywords used are not exclusive, for example, "predicting election results" is somewhat vague and does not represent the theme of either of the research questions.

Also, the databases used should be - 1. Emerald Insight, 2. Science Direct, 3. Scopus, 4. ACM, and 5. IEEE.

Validity of the findings

In the domain of election prediction using tweets, many researchers have used Facebook data as well. Hence validity of the study is somewhat compromised, if the research questions are wide enough to include "social media" domain.

Next, the results discussed in section 5 (Election Prediction Approaches) should be classified in two parts only, say, lexicon base approaches, or machine learning techniques, and/or both.This is done in the next section (6) where authors discussed trends in this field. This also suggest the overlap in the research questions which were not defined as per the norms in systematic review studies.

---

## Round 0.2 · Minor Revisions

Dear authors,

Your article has a few remaining issues. We encourage you to address the concerns and criticisms of the reviewers and resubmit your article once you have updated it accordingly.

Best wishes,

Reviewer 1 ·

Basic reporting

All requested changes in this section have been thoroughly validated.

Experimental design

All requested changes in this section have been thoroughly validated.

Validity of the findings

All requested changes in this section have been thoroughly validated.

Additional comments

All requested changes in this section have been thoroughly validated.

Reviewer 2 ·

Basic reporting

Thank you for considering the feedback provided to increase the quality of this manuscript. However, an updated list of changes is suggested below:


General
- I suggest Figure 1 and Figure 2 are re-created. Text should be aligned and the images shouldn't be blurry. Same feedback should be applied to Figure 6.
- I recommend replacing Figure 4 pie chart by a bar chart. Same feedback should be extended to Figure 5.
-Revise that all figures, equations and tables are cited in the text.
- I suggest that the references and citations are revised. For example, there are duplicates: Wang, H., Can, D., Kazemzadeh, A., Bar, F., and Narayanan, S. (2012). A system for real-time Twitter sentiment analysis of 2012 US presidential election cycle.

- I would recommend that the authors professionally proofread the paper carefully to ensure that it is free of language and formatting errors. This will improve the clarity and readability of the paper and help to ensure that the authors' ideas and findings are presented in the best possible light.


Results and Discussion
- In my humble opinion, this section is weak. I believe the results are embedded in the Methodology section. I suggest you split the theory, which should be part of the Methodology section, from the specific results to be part of the Results and discussion.
- I suggest you rename "Results and discussion" by "Findings", as a way to integrate both. In this section,
- The section needs further clarifications. You should comment on the main results of this research and compare those findings against the most relevant works in academia in the field. You should also present the main implications derived from this study from a theoretical, managerial and practical perspectives.
Conclusions

Conclusions
- The Conclusions section should be updated. This section should be revisited to clearly present the contributions of this manuscript to the existing literature. It is important to elaborate more on the justification about the limitations of the study. I suggest "Limitations and Future Directions" is a subsection of Conclusion.

Experimental design

Check the comments above

Validity of the findings

Check the comments above

·

Basic reporting

After reading the modified manuscript, it seems the authors have improved and incorporated the comments by the reviewers. Technically, the study is now improved. However, the smooth linking of the ideas, and reading-comforts for the readers, still need to be improved The figures captions require language corrections. .Some of the references also need to be carefully incorporated correctly, for example, reference [2].

I appreciate the efforts put by the authors to improve the readability and incorporating the suggested comments.

Experimental design

The study design is now more appropriate, the research questions are trimmed down to more precise phrases.. Although, the motivation is still vague and need more clear classification of the two era (like defining the two era in a more criteria based methodology, since AI started as early as 2012, though the focus still did not include the domain of sentiment analysis till 2014-15) should be 2008-2015 and 2015-2023. I would suggest to develop criteria to define the two distinct era.

Validity of the findings

Also, the findings (based on the developed criterion to classify the era) should be fine-tuned accordingly. As of now, the distinction between the two classes (era) is not very clear.

Additional comments

With these suggested minor corrections, the paper is now publishable. I would recommend to do one more extensive update of the manuscript before getting it published.

---

## Round 0.3 · accepted · Accept

Dear Authors,

Thank you for the hard work. Your paper was accepted following the third review process.

Best wishes,

Reviewer 2 ·

Basic reporting

I appreciate the effort put into implementing the requested changes. These changes have greatly contributed to the enhancement and solidification of the academic foundation of this manuscript. The research is well-executed and the writing is straightforward and precise. Therefore, I recommend that the paper be accepted

Experimental design

I appreciate the effort put into implementing the requested changes. These changes have greatly contributed to the enhancement and solidification of the academic foundation of this manuscript. The research is well-executed and the writing is straightforward and precise. Therefore, I recommend that the paper be accepted

Validity of the findings

I appreciate the effort put into implementing the requested changes. These changes have greatly contributed to the enhancement and solidification of the academic foundation of this manuscript. The research is well-executed and the writing is straightforward and precise. Therefore, I recommend that the paper be accepted

Additional comments

I appreciate the effort put into implementing the requested changes. These changes have greatly contributed to the enhancement and solidification of the academic foundation of this manuscript. The research is well-executed and the writing is straightforward and precise. Therefore, I recommend that the paper be accepted

·

Basic reporting

The manuscript is more readable with clarity of the steps and results. All the suggested review comments have been incorporated by the authors.

Experimental design

Authors have incorporated all the review comments in this section.

Validity of the findings

Authors have incorporated all the review comments in this section.

Additional comments

Authors have incorporated all the review comments in this section.